# Anti-Staphylococcal Activity of the Auranofin Analogue Bearing Acetylcysteine in Place of the Thiosugar: An Experimental and Theoretical Investigation

**DOI:** 10.3390/molecules27082578

**Published:** 2022-04-16

**Authors:** Lorenzo Chiaverini, Alessandro Pratesi, Damiano Cirri, Arianna Nardinocchi, Iogann Tolbatov, Alessandro Marrone, Mariagrazia Di Luca, Tiziano Marzo, Diego La Mendola

**Affiliations:** 1Department of Pharmacy, University of Pisa, Via Bonanno Pisano, 6, 56126 Pisa, Italy; lorenzo.chiaverini@farm.unipi.it (L.C.); diego.lamendola@unipi.it (D.L.M.); 2Department of Chemistry and Industrial Chemistry (DCCI), University of Pisa, Via G. Moruzzi, 13, 56124 Pisa, Italy; alessandro.pratesi@unipi.it (A.P.); damiano.cirri@dcci.unipi.it (D.C.); 3Department of Biology, University of Pisa, Via San Zeno 35–39, 56100 Pisa, Italy; a.nardinocchi@studenti.unipi.it; 4Institute of Chemical Research of Catalonia (ICIQ), The Barcelona Institute of Science and Technology, 43007 Tarragona, Spain; 5Dipartimento di Farmacia, Università degli Studi “G. D’Annunzio” Chieti-Pescara, Via dei Vestini, 66100 Chieti, Italy; amarrone@unich.it

**Keywords:** auranofin, gold, antibacterial agents, *S. aureus*, *S. epidermidis*, antibiotic resistance, metal-based drugs, acetylcysteine, ESI-MS, computational chemistry

## Abstract

Auranofin (AF, hereafter) is an orally administered chrysotherapeutic agent approved for the treatment of rheumatoid arthritis that is being repurposed for various indications including bacterial infections. Its likely mode of action involves the impairment of the TrxR system through the binding of the pharmacophoric cation [AuPEt_3_]^+^. Accordingly, a reliable strategy to expand the medicinal profile of AF is the replacement of the thiosugar moiety with different ligands. Herein, we aimed to prepare the AF analogue bearing the acetylcysteine ligand (AF-AcCys, hereafter) and characterize its anti-staphylococcal activity. Biological studies revealed that AF-AcCys retains an antibacterial effect superimposable with that of AF against *Staphylococcus aureus*, whereas it is about 20 times less effective against *Staphylococcus epidermidis*. Bioinorganic studies confirmed that upon incubation with human serum albumin, AF-AcCys, similarly to AF, induced protein metalation through the [AuPEt_3_]^+^ fragment. Additionally, AF-AcCys appeared capable of binding the dodecapeptide Ac-SGGDILQSGCUG-NH_2_, corresponding to the tryptic C-terminal fragment (488–499) of hTrxR. To shed light on the pharmacological differences between AF and AF-AcCys, we carried out a comparative experimental stability study and a theoretical estimation of bond dissociation energies, unveiling the higher strength of the Au–S bond in AF-AcCys. From the results, it emerged that the lower lipophilicity of AF-AcCys with respect to AF could be a key feature for its different antibacterial activity. The differences and similarities between AF and AF-AcCys are discussed, alongside the opportunities and consequences that chemical structure modifications imply.

## 1. Introduction

Antibiotic resistance is one of the biggest threats to global health. More than 1.2 million people died in 2019 as a direct result of antibiotic-resistant bacterial infections, according to the most comprehensive estimate to date of the global impact of antimicrobial resistance [1]. *Staphylococcus aureus* is the second among the most responsible bacteria for the fatal antibiotic resistance burden occurring in high-income countries [1,2]. In particular, methicillin-resistant *S. aureus* and *Staphylococcus epidermidis* are involved in a plethora of infections, such as hospital-acquired infections associated with high morbidity and mortality. Therefore, new antimicrobials, also including repurposed drug, to treat infections caused by antibiotic-resistant staphylococcal strains represent an urgent need.

In this frame, inorganic drugs represent a suitable option for the implementation of antibacterial agents capable to overcome the resistance phenomenon [3,4,5,6]. Indeed, metals and metal-based compounds have been used as antimicrobial agents since antiquity, on a merely empirical basis [7,8]. Similarly, various compounds bearing a metalloid center have been used or tested to fight infections [9,10]. 

Gold has emerged as a promising metal against bacterial infection. Among several gold(I)-based compounds, Auranofin is certainly the leading one [11,12]. Auranofin (1-thio-β-D-glucopyranosatotriethylphosphine gold-2,3,4,6-tetraacetate), marketed with the brand name Ridaura^®^, is in clinical use for the treatment of adults affected by rheumatoid arthritis. Beyond its approved use, in the last decades AF has been extensively repurposed for anticancer, antibacterial, antiviral and antiparasitic therapy, being also included in various clinical trials (see trials.gov website) [13,14,15,16].

Among the various indications for which AF has been repurposed, its use as antibacterial agent has turned out to be particularly promising. AF is highly effective against Gram-positive bacteria including multi-drug resistant pathogens, while it is substantially inactive against Gram-negative strains [17,18,19]. This different activity opened the way to various mechanistic hypotheses. AF is a metal-containing drug, and its overall pharmacological action likely relies on multiple interactions involving different mechanisms and biological substrates. Indeed, it is ascertained that its gold center can establish tight interactions with various selenol- or thiol-bearing proteins, with consequent impairment of key pathways important for cell survival. Accordingly, gold-based drugs are recognized to be efficient inhibitors of thiol redox homeostasis through direct blockage of the thioredoxin reductase enzyme [14,17,20,21]. However, other biosynthetic pathways have also been investigated for their role in concurring to the overall pharmacological activity of AF. Indeed, it has been proposed that other possible targets are the cell wall, DNA, and bacterial protein synthesis pathways [17]. Jackson-Rosario and Self, in 2009, reported an interesting mechanistic hypothesis for the antibacterial profile of AF. Specifically, these authors described the effects of AF against the oral pathogen *Treponema denticola*, suggesting that the observed effects were mediated by the inhibition of selenium metabolism which is crucial for the synthesis of selenoproteins [22]. Thus, the affinity of the gold center for selenium makes Auranofin highly reactive toward this element. The consequent decreased availability of Se for the synthesis of key bacterial proteins was determined to eventually lead to the inhibition of bacterial growth [14]. The lack of efficacy against Gram-negative strains is likely dependent on the permeability barrier conferred by the outer membrane, as extensively documented [18].

Importantly, the pharmacological action of AF is driven by the [AuPEt_3_]^+^ cation—representing the true pharmacophore—formed after the release of the thiosugar moiety in the biological environment. This cationic fragment is capable of binding to thiol- and selenol-containing proteins triggering a cellular response, eventually culminating in the desired pharmacological action [18,23,24].

A suitable strategy to modulate and improve the medicinal effects of AF is the replacement of the non-pharmacologically essential thiosugar with other ligands endowed with specific properties [14]. Based on this evidence, we prepared an AF derivative bearing the acetylcysteine ligand linearly coordinated to the gold(I) center (AF-AcCys, hereafter) Figure 1. 

The synthesis of this complex was previously reported; however, a comparative study on its antibacterial effects has never been carried out. Thus, our primary interest was directed to ascertain the retention of the antibacterial properties of AF toward representative bacterial strains and to confirm that, analogously to AF, even AF-AcCys is capable of [AuPEt_3_]^+^ release occurring with the simultaneous detachment of the N-Acetylcysteine ligand. This latter is a well-known drug included in the World Health Organization’s List of Essential Medicines (see the World Health Organization’s website). It is used in a variety of applications. It is effective in oral and intravenous forms as an antidote for acetaminophen poisoning but also for anti-inflammatory therapy for the treatment of cystic fibrosis patients [25,26]. According to this, assessing the antibacterial activity of AF-AcCys and its mechanistic aspects may open the avenue to the implementation of a dual Auranofin derivative. To this end, after AF-AcCys preparation, we carried out biological experiments to assess its antibacterial activity in comparison with AF. Interestingly, while the two compounds exert the same activity against *S. aureus*, a marked decrease in the activity of AF-AcCys was found against *S. epidermidis*. Bioinorganic experiments based on high-resolution mass spectrometry were carried out to investigate the mechanisms of AF-AcCys activation in the presence of human serum albumin (HSA) or a synthetic peptide mimicking the redox-active site (dTrxR(488–499)) of thioredoxin reductase. Next, the strength of the metal–ligand bonds for the two complexes was evaluated by means of quantum chemistry methods. Indeed, such approaches have been frequently and effectively employed for the description of the reactivity of metals and metallodrugs with biomolecules [27,28,29]. Overall, from this comparative evaluation, a nearly superimposable mode of activation of the two drugs emerged. However, some differences in the strength of the Au–S bond for AF and AF-AcCys as well as in their lipophilicity that might account for the strain-related observed activity were found.

## 2. Results and Discussion

### 2.1. Synthesis, Characterization, In-Solution Stability and LogP Evaluation of AF-AcCys

AF-AcCys was synthesized by modification of already reported methods, as described in the experimental section [30]. The complex was characterized through a multi-technique approach and its stability in the presence of DMSO and water was evaluated accordingly to an established method (see experimental section). 

The analysis of the obtained results revealed that AF-AcCys, similarly to AF, was very stable in our experimental conditions. In fact, from the NMR analysis, no significant changes in the spectral profiles emerged during the incubation, and the behavior of the two complexes was almost superimposable [18]. The ^31^P-NMR signal assigned to the phosphane ligand in the neutral complexes remained substantially unaltered even for long incubation times. In the case of AF-AcCys, alongside this signal, another very minor peak falling at 47 ppm attributable to the species [Au(PEt_3_)_2_]^+^ appeared also for very short incubation times [18]. However, this signal did not change over time, further supporting the high stability of the compound in the tested conditions (Appendix A). Additionally, the formation in solution of this species in a very low percentage is quite common for this family of compounds [18]. 

The evaluation of lipophilicity is mandatory for compounds that are considered potential new drugs. In fact, the lipophilic character of a given molecule is strictly related to its cellular accumulation and, then, to its pharmacological efficiency [31]. In this case, the LogP_o/w_ value for the AF-AcCys complex was determined through a modified shake-flask method using ICP-AES [23].

The value of the water/octanol partition coefficient determined for the studied compound was −0.95, while for the parent compound AF, a value of 1.6 was previously measured by some of us using the same procedure [32]. The results pointed to a significant increment of hydrophilicity that provides AF-AcCys with a better solubility in aqueous media with respect to AF, while maintaining a sufficient lipophilic character for the crossing of the cytoplasmic membrane [31,33].

### 2.2. Antibacterial Activity of AF-AcCys 

The in vitro activity of AF-AcCys and AF was evaluated against drug-resistant *S. aureus* and *S. epidermidis* strains. As shown in Table 1, the minimum inhibitory concentration (MIC) values of AF-AcCys versus *S. aureus* ranged from 0.25 to 0.5 µM. Similar results were obtained for AF versus the same strains. By contrast, higher MIC values (≥2) for AF-AcCys were observed when the molecule was tested versus *S. epidermidis*, while the same strains were susceptible to AF at lower MIC values (≤0.12 µM), suggesting that the addition of AcCys to AF might interfere with the antimicrobial activity of the pharmacophore versus *S. epidermidis* species.

### 2.3. Interaction Studies of AF-AcCys with HSA and dTrxR(488–499) Peptide

To study the mechanisms of activation and to confirm that AF-AcCys can interact with thiol-containing relevant biomolecules, the compound was incubated in the presence of human serum albumin (HSA). 

The high-resolution ESI mass spectrum recorded just after 2 h of incubation of AF-AcCys with HSA showed an almost complete protein metalation (Figure 2). In fact, as shown in Figure 2A, two main peaks were detected in the reference spectrum for HSA, one at 66,428 Da corresponding to the native free protein, and one at 66,557 Da attributed to post-translational modifications of serum albumin. Hence, the signal at 66,557 Da is perfectly in agreement with the presence of an additional cysteine residue bound to HSA via Cys34 [34,35]. This characteristic post-translational modification, i.e., cysteinylation, involving the unique cysteine residue present as a free thiol, is normally found in the serum-extracted protein [35,36]. After the incubation, the signal corresponding to the free protein was completely replaced by a new signal at 66,752 Da, corresponding to the monometallated protein with a [AuPEt_3_]^+^ fragment covalently bound on the Cys34 residue. Another small peak was also present at 66,914 Da, probably attributable to a whole molecule of AF-AcCys interacting with the protein.

Furthermore, it is interesting to notice how the peak corresponding to the cysteinylated form of the protein remained substantially unchanged, while for longer incubation times (24 h), the further increase of the metallated protein amount is consistent with the concomitant decrease of the signal at 66,557 Da (Appendix A).

Since the enzyme thioredoxin reductase (TrxR) is considered one of the main targets for cytotoxic gold compounds, the proof for a new compound to be able to interact with this biological target is of paramount importance [24,37,38]. To this aim, we selected a synthetic dodecapeptide sequence (i.e., Ac-SGGDILQSGCUG-NH_2_) corresponding to the C-terminal tryptic fragment (488–499) of human TrxR as a mimetic for the enzymatic active site [39,40]. This peptide possesses the peculiar –Cys–Sec– reactive motif, normally believed to be the possible binding site for gold(I)-containing molecules, and has been often used as a suitable model to describe the interaction of metallodrugs with TrxR [41,42]. After a very short time of incubation of AF-AcCys with the dTrxR(488–499) peptide (30 min), the resulting ESI mass spectrum clearly showed a signal at *m/z* 1497 indicating an interaction between the peptide and the gold-containing reactive fragment [AuPEt_3_]^+^ (see the SI for the spectra: Appendix A).

### 2.4. Computational Studies

In order to shed light on the effects determined by the ligand replacement, we performed DFT calculations of the energies required for breaking the bond between two unrelaxed molecular fragments, i.e., the snapping energies, and of both the bond-dissociation enthalpies and the free energies, i.e., BDE and BDFE, respectively, affecting the Au–P and Au–S bonds in both AF-AcCys and AF (Table 2). Indeed, the BDE values provide a better assessment of the energy cost of the dissociation compared to the BDFE values which also include –TΔS contributions. Above all, the energy cost for the breaking of the Au–P bond resulted to be upshifted by 3–6 kcal/mol (Table 2), thus indicating this bond is stronger than Au–S in both complexes. These computational data suggest that the AcCys and thiosugar moieties play the role of labile ligands in these metal scaffolds, whereas the phosphine fragments, tightly coordinated to the soft Au(I) metal centers, play the role of carrier ligands. In terms of either snapping energy or BDE, the energy cost for the dissociation of the Au–S bond in the AF-AcCys complex was detected to be lower by about 2 kcal/mol compared to that required in the case of AF (Table 2). The comparison of the BDFE values also showed that the Au–S bond is weaker in AF-AcCys than in AF, even though the estimated BDFE difference was only 1.0 kcal/mol (Table 2). In order to further rationalize our results, we performed the analysis of Mulliken charges (Appendix A). A slightly higher polarization of the Au–S bond in the AF-AcCys complex was estimated: indeed, the negative charge on S in the case of AF-AcCys was −0.56 compared to −0.51 in AF, while the positive charge on Au was +0.14 in AF-AcCys compared to +0.16 in AF.

This evidence suggests that the enhanced strength of the Au–S bond in Au-AcCys compared to AF is attributable to a stronger electrostatic contribution.

This kind of quantitative comparison of bond enthalpies and free energies is important because it can support experimental observations, allowing to assess if the strength of the labile Au–S bond represents the source of the disparity in the biological activity of the investigated complexes, among other possible chemical phenomena affecting their pharmacodynamics [29].

## 3. Conclusions

The selective modification of approved drugs to modulate their biochemical and pharmacological profiles is an interesting strategy potentially allowing the design of improved drugs [14]. In this frame, we were interested in replacing the thiosugar moiety of Auranofin with the acetylcysteine ligand. This modification allowed to obtain the AF-AcCys compound that was firstly characterized for its chemical features in comparison with Auranofin. It emerged that, similarly to Auranofin, AF-AcCys is stable in the presence of water and DMSO. However, a marked difference in the LogP value was found. Indeed, while AF has a value of 1.6, the AF-AcCys is far more hydrophilic, with a LogP value of −0.95. 

A detailed study of the reactivity behavior is compulsory if a synthesized molecule is intended to be used as a drug. Specifically, a gold-based compound like AF-AcCys endowed with antimicrobial activity against selected bacterial strains must necessarily undergo reactivity studies using some relevant target biomolecules. In our case, we selected one protein (HSA) and one synthetic dodecapeptide (dTrxR(488–499)). Serum albumin was selected because it is one of the most abundant proteins in the plasma, with a concentration of about 0.6 mM [43]. It plays a key role as a drug transporter in the bloodstream, especially for those molecules administered intravenously. The amino acidic chain of serum albumin contains 583 residues and presents 17 disulfide bonds and only one free cysteine residue (Cys34) that is commonly recognized as the preferential binding site for gold-containing complexes [35,44]. It has been already clarified that HSA is the major carrier for AF in the bloodstream [35], and the free and solvent-accessible thiolic group of Cys34 is the unique binding site due to its soft character, in agreement with the HSAB theory [44]. AF reacts with HSA through its pharmacophore moiety, i.e., the cation [AuPEt_3_]^+^, losing the thiosugar portion [23,45,46]. The data here showed that the reactivity of AF-AcCys is very similar to that of AF. Indeed, the new compound eagerly coordinates the protein through the cationic fragment [AuPEt_3_]^+^, likely at the Cys34 level, after the release of the L-Acetylcysteine ligand. An almost quantitative formation of the adduct -in terms of free thiol residues- already occurred after 2 h of incubation, with a substantial increase after 24 h due to the involvement of the cysteinylated Cys34 residues of HSA (Figure 2 and Appendix A).

The second biomolecule was chosen on the basis of the large amount of scientific literature that points to thioredoxin reductase as one of the main targets for bioactive gold-based compounds [37,47,48]. TrxR is critically involved in the regulation of intracellular redox metabolism, and targeting this enzyme has been regarded as a promising strategy for drug discovery [49]. For some years, some of us successfully employed a synthetic dodecapeptide reproducing the C-terminal tryptic fragment of TrxR as a valid reduced model for TrxR activity [39,40]. Hence, AF-AcCys was also challenged with this biomolecule and turned out to be able to rapidly interact with this target. Worthy of note is the extreme rapidity of the adduct formation that occurred after only 30 min of incubation. The mechanism of the binding is the same as that for the parent compound AF and analogous to that already described for HSA.

Theoretical calculations highlighted a slightly higher stability of the Au–S bond in AF-AcCys (bond dissociation enthalpy of 51.8 kcal/mol) in comparison with the Au–thiosugar bond in AF (49.8 kcal/mol). The greater strength of the Au–S bond in Au-AcCys originates from a stronger electrostatic interaction juxtaposed to the same bond in AF, as inferred from the Mulliken charges analysis. 

Overall, the results herein reported allow us to conclude that the replacement of the thiosugar moiety of Auranofin with acetylcysteine does not significantly affect the molecule’s activation profile, which implies the release of the acetylcysteine ligand with the formation of the pharmacologically active species [AuPEt_3_]^+^. 

Conversely, the higher hydrophilicity of AF-AcCys may account for the differential pharmacological activity towards the tested strains. Indeed, *S. aureus* is more hydrophilic than *S. epidermidis*. This difference in the surface characteristics is attributable to the different chemical structure of the microbial cell walls [50]. 

It is worth of mention that our results also support and integrate already obtained evidence. Indeed, it was previously reported that positively charged AF analogs have comparable antibacterial effects on both *S. aureus* or *S. epidermidis* [18]. In turn, this further confirms that what impacts the differential activity we observed should be the negative LogP value of AF-AcCys. Indeed, both AF and AF-AcCys are neutral [18,51]. In other words, the combination of the strong hydrophilic character of the therapeutic agent and its neutral charge seems indispensable for the antibacterial activity toward *S. aureus*. 

Our findings also indicate the [AuPEt_3_]^+^ moiety as the “true pharmacophore” in this family of complexes, the antimicrobial activity of which probably arises from the coordination of gold to specific targets after ligand exchange. 

As a consequence, with selective changes in the chemical structures of Auranofin, it is possible to retain the antimicrobial activity, coupling it with additional properties by choosing specific ligands. Finally, we provided hints on how structural modifications may selectively drive the drug activity toward specific bacterial strains, enhancing its specificity. 

## 4. Materials and Methods

### 4.1. General Remarks

All the solvents and reagents were purchased from Merck and used without further purification. NMR spectra were recorded at 293 K on a Bruker Avance II 400 MHz; chemical shifts (expressed in parts per million) were referenced to residual solvent peaks.

### 4.2. Synthesis and Characterization of AF-AcCys

The complex was synthesized by modification of already reported methods [52,53]. To L-Acetylcysteine (38 mg, 0.23 mmol) in ethanol (3 mL) we added (triethylphosphine)gold chloride (80 mg, 0.23 mmol) and a solution of sodium hydroxide in water (9 mg, 0.23 mmol, in 0.5 mL of water), under nitrogen atmosphere. The reaction mixture was magnetically stirred overnight. The colorless solution was filtered using celite in order to remove the precipitate and then evaporated under reduced pressure, obtaining a sticky white oil. The product was washed several times using diethyl ether, finally obtaining AF-AcCys as a white powder. Yield: 97 mg (88%). (see the Appendix A)

^1^H-NMR (400 MHz; DMSO-*d_6_*) δ: 12.43 (s; br; ^1^H; -COOH); 7.93 (d; 1H; *J*_HH_ = 7.8 Hz; -NH); 4.26 (m; 1H; -CH); 3.11 (dd; 1H; *J*_HH_ = 12.8, 5.2 Hz; -CH_2_); 2,97 (dd; 1H; *J*_HH_ = 12.7, 7.4 Hz; -CH_2_); 1.87 (m; 9H; -CH_3_ Ac. Cyst; -CH_2_ phosphine); 1.13 (m; 9H; -CH_3_ phosphine).

^13^C{^1^H}-NMR (100 MHz; DMSO-*d_6_*) δ = 172.94 (COOH); 169.56 (-CO); 57.66 (C-NH); 29.57 (CH_2_-S); 22.99 (CH_3_); 17.46 (d; *J*_CP_ = 31.4 Hz; -CH_2_ phosphine); 9.38 (-CH_3_ phosphine). 

^31^P{^1^H}-NMR (160 MHz; DMSO-*d_6_*) δ = 46.71 ([Au(PEt_3_)_2_]^+^); 36.67 (AF-AcCys).

Elemental analysis (CHN) found: C, 23.86; H, 4.05; N, 2.33. C_11_H_25_AuClNNaO_4_PS requires: C, 23.86; H, 4.55; N, 2.53.

### 4.3. Stability Study and LogP Determination of AF-AcCys and Auranofin 

The stability of AF-AcCys was assessed by ^31^P{^1^H}-NMR experiments recorded at increasing time intervals. The compound (24 mg, 0.05 mmol) was solubilized in a 1:1 DMSO/D_2_O mixture, and NMR spectra were recorded after 15 min, 2 h, 24 h and 48 h of incubation at 298 K. The stability of Auranofin was assessed by ^31^P{^1^H}-NMR experiments recorded at increasing time intervals. The compound (5 mg, 0.007 mmol) was solubilized in a 2:1 DMSO/D_2_O mixture (because of its lower solubility in aqueous media), and NMR spectra were recorded after 15 min, 2 h, 24 h and 48 h of incubation at 298 K. 

The LogP_o/w_ partition coefficient was determined by a modified shake-flask method [54]. Milli-Q water (250 mL) and n-octanol (250 mL) were shaken together for 72 h to allow saturation of both phases in a 500 mL flask. The suspension was allowed to separate at least for one week in the dark. A solution of the complex was prepared using the water phase (1 mM), and an equal volume of octanol was added. Biphasic solutions were mixed for 10 minutes and then centrifuged at 25 °C for 5 min at 6000 rpm to allow separation. Concentrations in both phases were determined through ICP-AES, following an already published mineralization protocol [55]. The reported log LogP value is defined as log10([complex]_oct_/[complex]_wat_). The final value was reported as the mean of three different determinations.

### 4.4. Interaction with HSA and dTrxR(488–499) Peptide

Stock solutions of HSA and dTrxR(488–499) peptide 10^−3^ M were prepared by dissolving the biomolecule in LC-MS-grade water. The stock solution 10^−2^ M of AF-AcCys was prepared by dissolving the compound in DMSO. Aliquots of the biomolecules stock solutions were mixed with aliquots of the gold compound stock solution at a biomolecule-to-metal ratio of 1:2 and diluted with an ammonium acetate solution 2 × 10^−3^ M (pH 6.8) to a final biomolecule concentration of 10^−4^ M. The protein-containing mixtures were incubated at 37 °C for 2 h and 24 h, while the peptide-containing sample was incubated at 37 °C for 30 min.

After the incubation, all solutions were sampled and diluted to a final biomolecule concentration of 5 × 10^−7^ M using a 2 × 10^−3^ M ammonium acetate solution, pH 6.8. In the final solutions, 0.1% *v*/*v* of formic acid was also added just before infusion in the mass spectrometer in order to enhance the ionization in the positive ion mode.

The ESI mass spectra were acquired on a TripleTOF^®^ 5600^+^ high-resolution mass spectrometer (AB Sciex, Framingham, MA, USA), equipped with a DuoSpray^®^ interface operating with an ESI probe. All spectra were acquired through direct infusion into the spectrometer at 5 μL/min of flow rate. The general ESI source parameters optimized for each biomolecule were as follows:
-HSA: positive polarity, ion spray voltage floating 5500 V, temperature 25 °C, ion source Gas 1 (GS1) 45 L/min; ion source Gas 2 (GS2) 0 L/min; curtain gas (CUR) 15 L/min, collision energy (CE) 10 V; declustering potential (DP) 200 V, acquisition range m/z 1000–2600. -dTrxR(488–499) peptide: positive polarity, ion spray voltage floating 5500 V, temperature 100 °C, ion source Gas 1 (GS1) 25 L/min; ion source Gas 2 (GS2) 25 L/min; CUR 20 L/min, CE 10 V; DP 50 V, acquisition range m/z 1000–2000.

For acquisition, Analyst TF software 1.7.1 (Sciex) was used, and deconvoluted spectra were obtained by using the Bio Tool Kit micro-application v.2.2 embedded in PeakView^TM^ software v.2.2 (Sciex).

### 4.5. Antibacterial Activity of AF-AcCys

Tests of antimicrobial activity of AF and AF-AcCys versus three clinical strains of either *S. aureus* or *S. epidermidis* (S.e2, S.e6, S.e7), isolated from device-associated infections, were carried out using the standard broth microdilution assay in 96-well plates, as recommended by the Clinical and Laboratory Standards Institute (CLSI) [56] Briefly, an aliquot of 250 μL of bacteria from −80 °C stocks was pre-inoculated in 5 mL of fresh Mueller Hinton broth (MHB) and incubated overnight shaking at 37 °C. Then, 250 μL from overnight growth cultures was inoculated in 5 mL of fresh MHB. When the culture reached the exponential log phase, the bacteria were diluted up to 10^6^ CFU/mL. Suspensions containing 10^6^ CFU/mL bacteria and two-fold serial dilutions of AF and AF-AcCys (from 2 to 0.125 µM) were incubated for 24 h at 37 °C. After that, bacterial growth was detected with the naked eye as turbidity in the well, according to CLSI guidelines [56]. The minimum inhibitory concentration (MIC) value was defined as the lower concentration of the antimicrobial substance capable of inhibiting bacterial growth.

### 4.6. Theoretical Calculations 

All DFT calculations were carried out with the Gaussian 16 C.01 quantum chemistry package [57]. Indeed, the DFT level of calculations is widely used for the structural analysis of metal complexes [58,59,60]. All geometrical optimizations, as well as single-point electronic and solvation energy calculations, were performed with the def2TZVP basis set [61] in water (IEFPCM) [62]. The range-corrected ωB97X hybrid density functional [63] was employed, since it gives a good description of the geometries and reaction profiles for transition-metal-containing compounds [64,65]. The utilization of the frequency calculations allowed the verification of the veracious nature of the stationary points and to accurately gauge zero-point energy (ZPE) and thermal corrections to thermodynamic properties. It was recently demonstrated that the employed IEFPCM continuum solvent method yields substantially smaller errors than other continuum models for aqueous free energies of solvation for neutrals, cations and anions and is especially efficacious for the calculations of solution properties necessitating accurate estimation of solution free energies [66]. The strength of either the Au–P or the Au–S bond within the investigated complexes was evaluated in terms of snapping energy, bond-dissociation enthalpies (BDE) and bond-dissociation Gibbs free energies (BDFE). Snapping energies were calculated by subtracting the single point energies of the two bonding fragments (at the geometry assumed in the complex) from the electronic energy of the optimized complex. BDE/BDFE are the differences between the enthalpy/free energy of the fully optimized complexes and the enthalpies/free energies of the bonding fragments after relaxation.

## Figures and Tables

**Figure 1 molecules-27-02578-f001:**
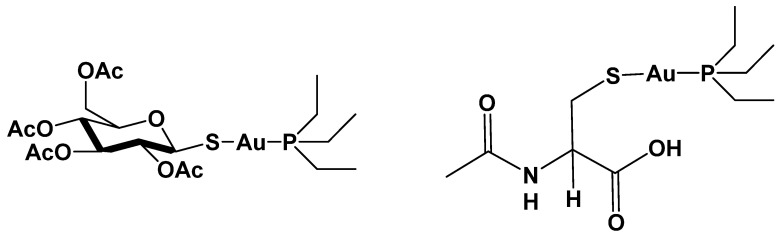
Chemical structure of Auranofin (AF, **left**) and its analogue bearing L-Acetylcysteine in place of the thiosugar (AF-AcCys, **right**).

**Figure 2 molecules-27-02578-f002:**
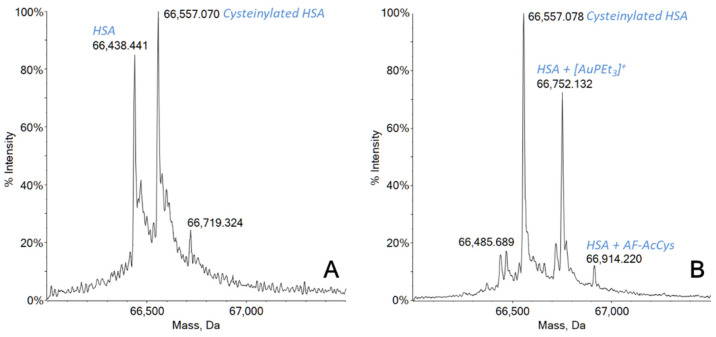
Deconvoluted ESI mass spectra of (**A**) HSA 5 × 10^−6^ M in 20 mM ammonium acetate solution, pH = 6.8, and (**B**) HSA 5 × 10^−6^ M incubated with AF-AcCys (1:2 ratio) for 2 h at 37 °C in 20 mM ammonium acetate solution, pH = 6.8.

**Table 1 molecules-27-02578-t001:** MIC (µM) values of AF-AcCys and AF.

ID Strain	Origin	Resistance Profile ^b^	AF-AcCys (µM)	AF (µM)
*S. aureus* 10	pacemaker	OXA; MET; LVX; PEN	0.5	0.5
*S. aureus* 11	osteomyelitis	ERY; CLI; PEN	0.25	0.25
*S. aureus* 12	CVC ^a^	ERY; PEN	0.5	0.5
*S. epidermidis* 2	emocolture	ERY; GEN; FA; OXA; MET; TET; CLI; LVX;	2	≤0.12
*S. epidermidis* 6	CVC	GEN; OXA; MET; TET; LVX;	2	≤0.12
*S. epidermidis* 7	pacemaker	ERY; GEN; OXA; MET; CLI; LVX; TMP/SMX	>2	≤0.12

^a^ CVC: Central Venous Catheter; ^b^ OXA: oxacillin; MET: methicillin; LVX: levofloxacin; PEN: penicillin; ERY: erythromycin; CLI: clindamycin; GEN: gentamycin; FA: fusidic acid; TET: tetracycline; TMP/SMX: trimethoprim/sulfamethoxazole.

**Table 2 molecules-27-02578-t002:** Snapping energies, bond-dissociation enthalpies (BDE) and bond-dissociation Gibbs free energies (BDFE). All values are in kcal/mol.

Complex	Bond	Snapping Energies	BDE	BDFE
AF-AcCys	Au-S	54.8	51.8	40.0
	Au-P	58.8	54.4	42.2
AF	Au-S	52.6	49.8	39.0
	Au-P	59.1	54.7	43.7

## Data Availability

Not applicable.

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
