# Peer review of "Anti-Staphylococcal Activity of the Auranofin Analogue Bearing Acetylcysteine in Place of the Thiosugar: An Experimental and Theoretical Investigation"

_molecules, 2022, doi:10.3390/molecules27082578_

Round 1

Reviewer 1 Report

The authors aimed their work to summarize Antimicrobial activity of the Auranofin analogue bearing acetylcysteine in place of thiosugar against Staphylococcus strains. Topic is of much interest. Overall work is interesting and very well written. I have some major comments. If these corrections are made, I believe the above work can be published to Antibiotics.

  1. Title: Need to change Anti-staphylococci activity of the Auranofin analogue bearing acetylcysteine in place of thiosugar: An experimental and theoretical investigation). Due to authors just study activity against Staphylococcus strains.
  2. A lot of typo and duplicate full stop, Why?
  3. where is the objective of the article? I try to find it in abstract and at end of introduction?.
  4. Abstract: need to rewrite to show results. e.x: line 27-30 long sentence, please correct it.
  5. Please kindly ensure that scientific names are italicized.
  6. Use S. aureus and S. epidermidis after you've used their complete names the first time.
  7. In table 1: This data would be much better if you included a column for the origins of the isolates, as well as a resistance profile for those isolates. You may also talk about your resitance profile in the context of the Discussion.
  8. Line 377-388: How you measure??? The bacterial growth was detected with naked-eye as turbidity in the well. The minimum inhibitory concentration (MIC) value was defined as the lower concentration of the antimicrobial substance capable of inhibiting bacterial growth. In order to make things easier, you'll need to use a quantitative approach. Add a picture of the 96-well plate to illustrate the anti-staphylococcal activity in microdilution plate. This may be another solution.
  9. Please separate Discussion and conclusions. Add conclusions at the end of manuscript.
  10. Improve the discussions: 

    I should have expected the authors to discuss elaborately their findings as compared with other similar studies with antistaphylococcal activity of Auranofin. It is important to build on the discussion to highlight the originality of the work.

Reviewer 2 Report

The manuscript is well written. The title, abstract, scheme, tables and figures of the manuscript are adequate to the content. The experimental part gives enough details about the synthesis and spectral data. 

However the paragraphs titles are wrong:

2. Results and Results and Discussion -> 2. Results and Discussion

3. Discussion -> 3. Conclusion 

The NMR spectra aren’t fully described and this can call into question the overall structural elucidation of the obtaining compounds. Please provide copy of NMR spectra in the supporting documents.

Round 2

Reviewer 1 Report

All my concerns have been taken in consideration.